# Outpatient Nephrotoxic Medication Prescription after Pediatric Intensive Care Acute Kidney Injury

**DOI:** 10.3390/children8110948

**Published:** 2021-10-21

**Authors:** Claire Lefebvre, Marc Dorais, Erin Hessey, Michael Zappitelli

**Affiliations:** 1Department of Pediatrics, Centre Hospitalier Universitaire Sainte-Justine, Montreal, QC H3T 1C5, Canada; 2Department of Epidemiology, Biostatistics, and Occupational Health, McGill University, Montreal, QC H3A 1A2, Canada; 3StatSciences Inc., Notre-Dame-de-l’Île-Perrot, QC J7V 7P2, Canada; marc.dorais.statsciences@gmail.com; 4Faculty of Medicine and Dentistry, University of Alberta, Edmonton, AB T6G 2R3, Canada; hessey@ualberta.ca; 5Department of Pediatrics, Division of Nephrology, Toronto Hospital for Sick Children, Toronto, ON M5G 1X8, Canada; 6Department of Pediatrics, Division of Nephrology, McGill University, Montreal, QC H3A 1A2, Canada

**Keywords:** pediatrics, renal, kidney disease, nephrotoxicity, prescription patterns

## Abstract

Background: Nephrotoxic medication (NTM) avoidance may prevent further kidney damage in children with acute kidney injury (AKI). We compared outpatient NTM prescriptions in children with or without AKI during pediatric intensive care (PICU) hospitalization. We hypothesize that children with AKI are prescribed NTMs at the same rate as those without it. Methods: This was a retrospective administrative data study of children <18 years, admitted to two PICUs in Montreal, Canada, from 2003 to 2005, with ≥30 days of provincial drug coverage. We evaluated the presence of ≥3 outpatient NTM prescriptions during the first year and 5 years after PICU discharge. Results: Of 970 children, 23% had PICU AKI. In the 1st–5th years after discharge, 18% AKI vs. 10% non-AKI and 13% AKI vs. 4% non-AKI patients received ≥3 NTM prescriptions, respectively. There was no association between PICU AKI and prescription of ≥3 NTMs during the first year (adjusted RR 1.02 [95% CI 0.95–1.10]) nor in the first 5 years post-discharge (adjusted RR 1.04 [95%CI 0.96–1.12]). Conclusions: By offering a better understanding of the current state of outpatient NTM prescription to children with AKI, our study is a step toward considering strategies such as knowledge translation interventions for decreasing NTM exposure and improving outcomes in children with AKI.

## 1. Introduction

Acute kidney injury (AKI) is an important public health concern. AKI or abrupt kidney dysfunction occurs in ~20% of children in the pediatric intensive care unit (PICU) and <5% of all hospitalized children [1]. While its etiology is commonly multifactorial in the hospital setting, AKI can occur and be exacerbated due to administration of nephrotoxic medications [2,3,4]. There is mounting evidence that AKI may be associated with the development of subsequent chronic kidney disease (CKD) [5,6], which is a potent cardiovascular disease risk factor [7,8].

There are no standardized guidelines on how to follow children with AKI after hospital discharge. Recent work suggests that many children with AKI are not followed for the possible development of CKD, suggesting that primary care providers may not be aware of the potential long-term effects of AKI [9]. An important concept in managing AKI and CKD is preventing further kidney damage, including nephrotoxic medication avoidance. If AKI is a risk factor for long-term CKD development, ideally, children with a history of AKI would not receive nephrotoxic medications, unless necessary. Nephrotoxic medication avoidance after AKI represents an opportunity for process of care and education interventions to mitigate the effects of AKI on long-term kidney function. We recently showed in a large UK cohort that children with CKD are often prescribed nephrotoxic medication by primary care providers [10]. Whether healthcare providers alter nephrotoxic medication prescriptions based on the history of AKI in children remains completely unknown.

Using administrative healthcare data, our objective was to describe and compare outpatient prescriptions of nephrotoxic medications after a PICU hospitalization in children who did vs. children who did not develop AKI during PICU admission. We hypothesized that due to the under-recognition of the potential importance of AKI in long-term kidney outcomes, children who developed AKI during PICU admission would be prescribed nephrotoxic medications after discharge at a rate similar to children who did not develop AKI.

## 2. Materials and Methods

### 2.1. Design, Setting, and Patient Selection

This is a secondary analysis of a previously published retrospective cohort study including children <18 years old with a valid provincial health card number, who were admitted to the PICU of the Montreal Children’s Hospital (MCH) (Montreal, Quebec, Canada) or the Centre Hospitalier Universitaire (CHU) Sainte-Justine (Montreal, Quebec, Canada) between 1 January 2003 and 31 March 2005, inclusively. Children with end-stage kidney disease (ESKD, including kidney transplant) before the index admission were excluded [11]. For this analysis, only the following children were included: survived to the index hospital discharge; registered to the Régie de l’assurance maladie du Québec (RAMQ, described below) provincial drug plan for at least 30 days after index hospitalization discharge. We only included the first PICU-associated hospitalization during the study period. (Figure 1). The follow-up start date was the date of discharge from the index hospitalization, and follow-up continued for up to 7 years after hospital discharge, death, or patient’s exit from the provincial drug plan for ≥6 months. Approvals from institutional research ethics boards and the Commission d’Accès à l’Information du Québec (provincial data monitoring board) were obtained.

### 2.2. Data Collection and Sources

Index hospitalization and PICU data were collected by a retrospective chart review on standardized forms which were pre-tested for reliability [12]. These variables, previously described, included: sociodemographic data; pre-PICU patient characteristics (comorbidities, baseline kidney function); PICU admission variables (e.g., primary diagnosis; Pediatric Risk of Mortality (PRISM) score [13]; daily medication (including vasopressors, diuretics, antibiotics, non-steroidal anti-inflammatories; steroids), duration of mechanical ventilation, daily serum creatinine (SCr), and urine output); and hospital outcomes (e.g., length of stay).

Chart data were merged with data from the Quebec Vital Statistics Registry (to evaluate post-discharge mortality) and administrative healthcare databases of the RAMQ and Med-Echo (collectively including administrative healthcare data from 1 year before the first PICU admission in the cohort during the study period to 5 years after the last admission during the study period) [11]. The administrative healthcare databases provided data on healthcare encounters (e.g., hospitalizations, physician visits), demographic and diagnosis/procedure codes (using the International Classification of Diseases—9 and 10) data (used to calculate the social and material deprivation indices as a marker of socio-economic status as well as the Pediatric Complexity Algorithm to reflect patient health complexity) [14,15]. Outpatient medication prescription data are included in the RAMQ databases.

Quebec has a single-payer health insurance plan. However, medications are reimbursed by a separate public drug plan which covers ~48% of the province’s children [16]. The RAMQ pharmaceutical services file provides information on medications reimbursed by the provincial insurer, including drug name, date the prescription was collected, and prescription dose, form, and duration. For children <18 years of age to be covered, their parents/guardians must be registered with the RAMQ drug plan. Patients are registered to the drug plan until they move out of a province or enroll in a private insurance plan (e.g., employer). We defined “continuous RAMQ medication coverage” as gaps in coverage <6 months.

### 2.3. AKI Definition

AKI during PICU hospitalization was defined from retrospective chart data using KDIGO (Kidney Disease: Improving Global Outcomes) criteria, classifying AKI based on the degree of SCr rise from baseline or the urine output decrease [17]. For patients with SCr measured in the three months before index PICU admission, the lowest SCr was the baseline SCr [17]. For patients with no baseline SCr available, we estimated it using the Chronic Kidney Disease in Children (CKiD) bedside equation (eGFR = 36.5 × (height in cm/SCr in µmol/L)) (1) to back-calculate baseline SCr, using patient height and assuming an age-based normative estimated glomerular filtration (eGFR) value (2) (for children ≤2 years) or an eGFR = 120 mL/min/1.73 m^2^ (for children >2 years) [18,19]. For patients with no height available, we used the height-independent Hoste equation [20] to back-calculate baseline SCr. Both equations have been validated in our population [21]. Patients for whom AKI could not be ascertained, because no SCr or urine output was recorded during PICU stay, were assumed not to have AKI [22]. The rationale behind this is that children without SCr or urine output measured during PICU are unlikely to be severely ill and therefore unlikely to have developed AKI. We previously showed that patients for whom AKI ascertainment data were not available were extremely similar in characteristics to patients with no AKI [23]. As a secondary analysis, we evaluated the presence of Stage 2 AKI or worse (≥doubling of SCr from baseline or dialysis for AKI) during PICU (vs. not having AKI or Stage 1 AKI).

### 2.4. Outcome: Post-Discharge Outpatient Medication Prescriptions

Our definition of a nephrotoxic medication was derived from three recent studies using robust methodologies to generate lists of medications recognized as nephrotoxic in adults or children [24,25,26]. In addition to considering data from these studies, we assessed the face validity of the included medications. We reached consensus about the inclusion of additional drugs that did not appear on these lists but were deemed to have sufficient evidence supporting their nephrotoxicity. We then developed two lists: one of established nephrotoxic medications and another of potential nephrotoxic medications (Appendix A). Established nephrotoxic medications included those considered nephrotoxic in at least two of the three reference studies. To this list, we added medications from the following classes: non-steroidal anti-inflammatory drugs (NSAIDs), aminoglycoside antibiotics, proton pump inhibitors, and salicylates. Our second list was broadened to include medications considered “potentially” nephrotoxic (providing as wide a capture of potential nephrotoxic medications as possible). This list included our established nephrotoxic medications plus medications considered to have nephrotoxic “potential” in any of the three reference studies or medications which only one of the three studies considered established nephrotoxic. Because of their possible indication in kidney disease following AKI, we considered angiotensin-converting enzyme (ACE) inhibitors to be potentially nephrotoxic in our study.

Our primary outcomes were: (1) presence (yes/no) of ≥3 established nephrotoxic medication prescriptions during the first year and during the first 5 years after PICU discharge and (2) receiving (yes/no) ≥3 established nephrotoxic medication prescriptions at 1-year intervals after PICU discharge. The rationale for using a cutoff of ≥3 medications was the association between this level of nephrotoxic medication exposure and incident AKI in hospitalized children [3,25]. Results were repeated for established and potential nephrotoxic medications.

Secondary outcomes included (a) presence of ≥3 potential nephrotoxic medication prescriptions and (b) presence of ≥3 established and potential nephrotoxic medication prescriptions from different classes (e.g., NSAIDS; antivirals). The secondary outcomes were evaluated during the first year and the first 5 years of follow-up.

### 2.5. Analysis

Analyses were performed using R, version 3.3.3 (R Foundation for Statistical Computing, Vienna, Austria) and SAS statistical software, release 9.4 (SAS Institute Inc., Cary, NC, USA). Characteristics were reported as appropriate based on distribution, and variables were compared between groups using distribution-appropriate univariable analyses (e.g., t-tests, Mann–Whitney U tests, Chi-square with or without Yate’s correction, Fisher’s exact test). Proportions of AKI vs. non-AKI patients who received ≥3 individual established nephrotoxic medication prescriptions after hospital discharge and at 1-year intervals, up to 5 years after discharge, were compared. Multivariable log binomial regression was used to calculate risk ratios (RR, 95% confidence interval [CI]) for the association between AKI (and also Stage 2/3 AKI) with receiving ≥3 established nephrotoxic medication prescriptions during the first year and in the first 5 years of follow-up. Similar analyses were performed to evaluate associations with our secondary outcomes (i.e., number of potential nephrotoxic medication prescriptions). The proportions of the most common classes of nephrotoxic medications prescribed by AKI-status group were calculated.

A priori covariates included in all multivariable models were age, gender, rural vs. urban home, pediatric medical complexity [15], and social and material deprivation indices [14]. For each multivariable model of our primary and secondary outcomes, additional covariates were selected if they had a significant association (*p* < 0.05) with both the exposure (AKI) and the specific outcome in univariable analyses. If data were missing for covariates, a complete case approach was used (i.e., patients with data on all relevant covariates were included in multivariable regressions. This resulted in <6% loss of subjects in all multivariable analyses). If there were too few observations within categorizations of a covariate, the covariate was excluded from the model. Cardiac surgery was explored as an effect modifier (planned a priori analysis) by including it as an interaction term with AKI in our multivariable model for our primary outcome (≥3 established nephrotoxic medication prescriptions in the first year of follow-up). In regression models evaluating outcomes at 1 and 5 years of follow-up, only patients with ongoing follow-up at those time intervals were included in the analyses.

## 3. Results

### 3.1. Cohort Description

There were 2499 patients hospitalized at the PICU from 2003 to 2005. Figure 1 shows reasons for exclusion, leading to an analysis cohort of 970 children (the majority excluded due to neonatal age and lack of RAMQ provincial drug plan coverage for ≥30 days). RAMQ drug plan coverage in our cohort closely resembled province-wide RAMQ coverage rates of 48% for children [16]. The characteristics of included and excluded children were similar, except that a higher proportion of the included children had elevated levels of social and material deprivation (Appendix A). Two hundred and twenty-three children (23%) developed AKI during PICU hospitalization (*n* = 85 (9%) with ≥Stage 2 AKI). Comparisons of characteristics between AKI and non-AKI patients are presented in Appendix A. Average follow-up time was 5.9 years. At 1 and 5 years of follow-up, 953 (98%) and 910 (94%) patients (no statistically significant difference between AKI groups), respectively, had continuous follow-up, defined as follow-up with less than a 6-month gap in RAMQ drug coverage.

### 3.2. Association of AKI with the Number of Nephrotoxic Medication Prescriptions

Figure 2 illustrates the proportions of patients with and without AKI who received ≥3 established and potential nephrotoxic medication prescriptions at yearly intervals after hospital discharge. During the first year, 18% of AKI and 10% of non-AKI patients received ≥3 established nephrotoxic medication prescriptions. In addition, 39% of AKI and 29% of non-AKI patients received ≥3 potential nephrotoxic medication prescriptions (*p* = 0.02). In the fifth year post-discharge, 13% of AKI and 4% of non-AKI patients received ≥3 established nephrotoxic medication prescriptions, whereas 22% of AKI and 12% of non-AKI patients received ≥3 potential nephrotoxic medication prescriptions (*p* < 0.01). When considering the entire 7 years of follow-up, 59% of AKI and 56% of non-AKI patients received ≥3 potential nephrotoxic medication prescriptions at some time (*p* = 0.58).

Table 1 shows that there was no association between AKI during PICU admission and prescription of ≥3 established nephrotoxic medications during the first year after hospital discharge (adjusted RR 1.02 [95% CI 0.95–1. 10], adjusted for variables shown in Table 1 legend) or in the 5 years after hospital discharge (adjusted RR 1.04 [95%CI 0.96–1.12], adjusted for variables shown in Table 1 legend). The presence of ≥Stage 2 AKI (vs. no AKI or Stage 1 AKI) was also not associated with the outcomes in adjusted analyses (Table 1).

Cardiac surgery was not an effect modifier of the AKI–outcome relation (i.e., did not alter the association between AKI and nephrotoxic medication prescription; interaction term *p* value = 0.4). In the 1- and 5-year outcome multivariable analyses, higher pediatric medical complexity was consistently associated with the presence of ≥3 established nephrotoxic medication prescriptions (*p* < 0.05 in adjusted models). When we considered potential nephrotoxic medications in our definition, adjusted associations between AKI in PICU with 1-year and 5-year presence of ≥3 potential nephrotoxic medication prescriptions were extremely similar (shown in Appendix A).

### 3.3. Association of AKI with the Number of Nephrotoxic Prescriptions from Different Classes

Figure 3 shows that 3% of AKI and <1% of non-AKI children received medications from ≥3 separate established nephrotoxic classes in 1 year after hospital discharge (*p* < 0.01); in the fifth year after discharge, no patient received medications from ≥3 potential nephrotoxic classes. Table 1 shows that there was no statistically significant association between AKI and prescription of ≥3 established nephrotoxic medication prescriptions from different classes (adjusted RR 1.02 [95% CI 0.97–1.08] in the first year, adjusted RR 1.02 [95% CI 0.96–1.09] in the first five years; bottom of Table 1, adjusted for variables indicated in the legend). These results were extremely similar when defining the exposure as the presence of ≥Stage 2 AKI (shown in Table 1).

### 3.4. Distribution of the Prescribed Nephrotoxic Medication Classes

When considering the full 7 years of available follow-up, the most commonly prescribed established or potential nephrotoxic medication class was beta-lactam antibiotics (23% of patients with AKI and 42% of patients without AKI) (Appendix A, respectively). ACE-inhibitors were the second most commonly prescribed medication class to patients with AKI (18% of AKI patients throughout the study period). Proton pump inhibitors represented another commonly prescribed medication class and were prescribed to 14% and 13% of AKI and non-AKI patients, respectively. NSAIDs were prescribed to 3% of all patients, regardless of their AKI status.

## 4. Discussion

In the first year after PICU discharge, the RR of receiving ≥3 nephrotoxic medication prescriptions for children with AKI did not differ substantially from that for children without AKI. Almost 20% of the patients with a documented episode of AKI were prescribed ≥3 established nephrotoxic medication prescriptions in the year after their discharge from the PICU compared to 10% of children without AKI. Three or more potential nephrotoxic medications were prescribed to just under 40% of children with AKI and to just under 30% of children without AKI in the year following PICU discharge.

These findings are important because children with AKI may be at risk for subsequent AKI and for CKD progression [5,27]. Avoiding nephrotoxic medications is important in this population, as it represents one of the few modifiable risk factors for kidney disease progression. Though individual nephrotoxic medications are sometimes inconsistently associated with AKI, in the pediatric non-critical care population, exposure to ≥3 nephrotoxic medications has been shown to nearly double the positive predictive value of developing AKI (compared to ≤2 nephrotoxic medications) and has been previously reported to be associated with a 30% AKI incidence [3,25]. The high burden of outpatient nephrotoxic medication exposure in our study suggests a need for increased awareness of this practice among discharging and treating physicians. A previous study examining healthcare utilization in our cohort’s patients noted that fewer than 25% of those patients with a PICU diagnosis of AKI saw a nephrologist in the 5 years after their PICU discharge [22]. This highlights the need for post-AKI follow-up guidelines for these children.

To our knowledge, this is the first study evaluating the prescription of outpatient nephrotoxic medication following AKI in a PICU population. Most studies in this population have focused on inpatient prescriptions. Existing hospital studies have exposed worrying trends in prescribing practices to children with AKI, as well as promising interventions to mitigate these trends. One study described the effects of an institution-wide SCr surveillance program which resulted in a dramatic drop in inappropriate prescriptions and in cases of AKI (38% and 64%, respectively) [28]. Hospital-based studies overlook an important aspect of the clinical care pathway of many AKI patients—the outpatient setting. This is often the most frequent point of contact these children have with health care and has inherent differences with respect to the hospital setting. Regular SCr surveillance is feasible in hospital and unlikely to result in an increased burden of care in children already undergoing regular blood tests; such interventions may be impractical in the outpatient setting. Educational campaigns focused on increasing awareness of the importance of nephrotoxic medication avoidance may be of value.

Several commonly prescribed nephrotoxic medications identified were medications known to be commonly used in children (e.g., beta-lactam antibiotics, proton-pump inhibitors) and may often be difficult to avoid. However, alternatives do exist, and we propose, as supported by our study, that in the context of avoiding the overuse of antibiotic prescriptions, considering potential adverse effects should play a role in decision making. ACE-inhibitors were commonly prescribed to patients with a history of AKI. This may reflect the high proportion of cardiac surgery patients for whom these medications may be indicated for afterload reduction and for the treatment of hypertension. However, considering their nephrotoxic potential, their widespread use in these at-risk populations should prompt the consideration of a close monitoring of kidney function.

Our study has limitations. This study was conducted in a region with universal health care, and the results may therefore not be applicable to other contexts. Our hospitalization data also dates to 2003–2005, and prescribing practices may have changed since then. We did not have the indication for nephrotoxic medication prescriptions and can therefore not comment directly on the justification for their use or whether safer alternatives could reasonably have been considered. Though our study details evidence of a high nephrotoxic burden in children with previous AKI, we cannot comment on the impact of prescribing nephrotoxic medication on the evolution of these children’s kidney disease or on AKI recurrence. Furthermore, the absence of SCr measurements in a large proportion of our cohort made the identification of AKI difficult in many patients. This is a common feature of many studies of pediatric AKI and highlights how important it is to record these parameters in all children admitted to the PICU. However, our decision to include urine output criteria as an inclusion parameter would have helped to offset this lack of sensitivity. Lastly, in our study, the proportion of patients exposed to ≥3 potential or established nephrotoxic medications decreased with each passing year after PICU discharge, which suggests that the initial prescriptions may have been a response to the acute illness leading to their PICU hospitalization. We attempted to account for disease severity in our models by controlling for pre-admission diagnoses and PICU factors, but there may have been residual confounding.

## 5. Conclusions

By offering a better understanding of the current state of nephrotoxic medication prescribing to children with AKI in the outpatient setting, our study is a step toward considering strategies such as knowledge translation interventions for decreasing nephrotoxic medication exposure and, ultimately, improving outcomes in children with AKI.

## Figures and Tables

**Figure 1 children-08-00948-f001:**
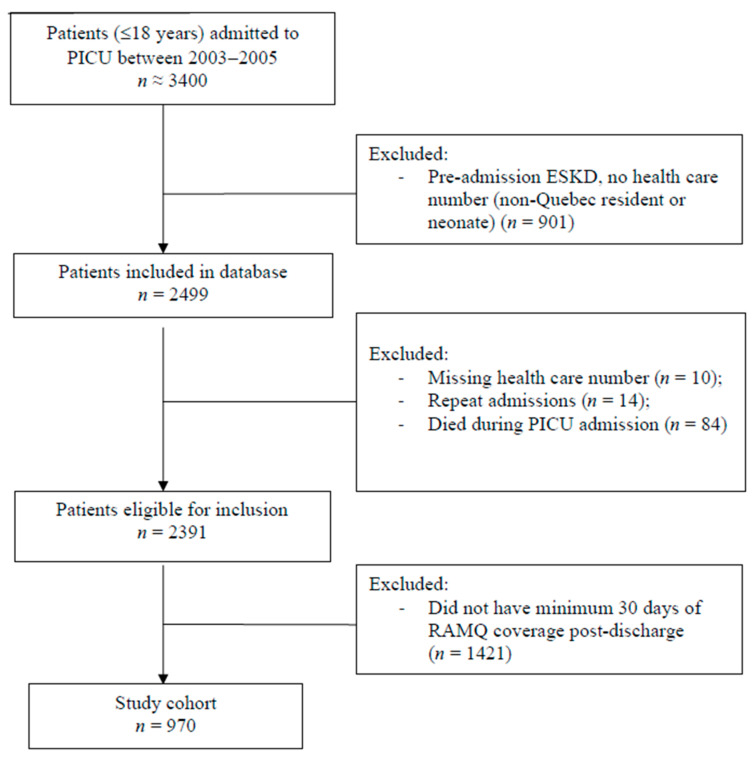
Flow Diagram of Patient Inclusion from the PICU and RAMQ Database. Abbreviations: PICU: pediatric intensive care unit; ESKD: end-stage kidney disease; RAMQ: Régie de l’assurance maladie du Québec.

**Figure 2 children-08-00948-f002:**
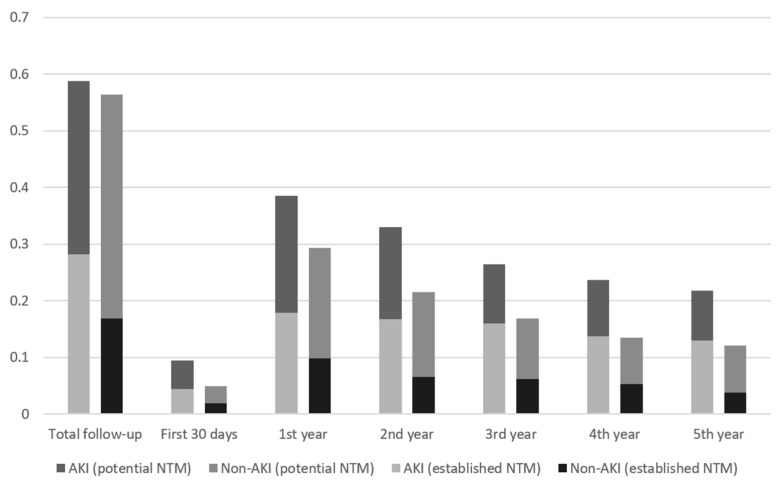
Percentages of AKI and non-AKI Patients Receiving ≥3 Nephrotoxic Medication Prescriptions by Year Since PICU Discharge. Bar chart of proportions of AKI and. Non-AKI patients who received ≥3 nephrotoxic medication prescriptions by year since PICU discharge. Numbers above figure bars represent the total number of participants with continuous follow-up at the time interval after discharge (denominator).

**Figure 3 children-08-00948-f003:**
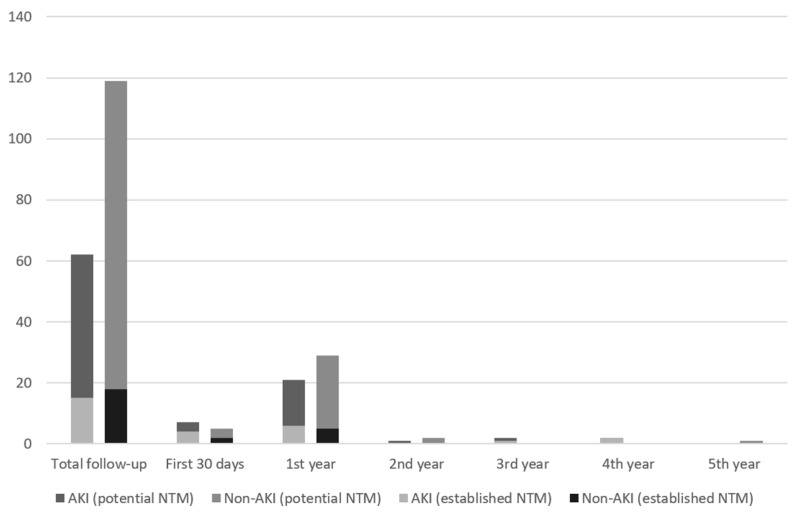
Percentages of AKI and non-AKI Patients Receiving ≥3 Nephrotoxic Medication Classes by Year Since PICU Discharge. Bar chart of proportions of AKI and Non-AKI patients who received ≥3 nephrotoxic medication prescriptions by year since PICU discharge. Numbers above bars represent the total number of participants with continuous follow-up at the time interval after discharge (denominator).

**Table 1 children-08-00948-t001:** Adjusted Association Between AKI and Administration of ≥3 Established Nephrotoxic Medication Prescriptions and Classes.

	AKI vs. no AKI		Stage 2/3 vs. no AKI or Stage 1 AKI	
	First Year (*n* = 953) Adjusted RR (95% CI)	First 5 Years (*n* = 910) Adjusted RR (95% CI)	First Year (*n* = 953) Adjusted RR (95% CI)	First 5 Years (*n* = 910) Adjusted RR (95% CI)
≥3 established nephrotoxic medication prescriptions	1.02 (0.95–1.10) ^a^	1.04 (0.96–1.12) ^b^	1.03 ^a^ (0.92–1.14)	1.04 (0.94–1.16) ^b^
≥3 established nephrotoxic medication classes	1.02 (0.97–1.08)	1.02 (0.96–1.09) ^c^	1.02 (0.91–1.15)	1.02 ^c^ (0.93–1.13)

Abbreviations: AKI: Acute Kidney Injury; RR: Risk Ratio. NB: All models include covariates: age, gender, rural versus urban home address, pediatric medical complexity algorithm, and social and material deprivation indices. Additional model covariates (associated with the exposure and outcome of the analysis) include: a cardiac surgery, trauma, diabetes, diuretics, antibiotics, PRISM score; b cardiac surgery, trauma, diuretics, antibiotics, PRISM score; c antibiotics.

## Data Availability

The data presented in this study are available on reasonable request from the corresponding author.

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
