# Peer review of "Outpatient Nephrotoxic Medication Prescription after Pediatric Intensive Care Acute Kidney Injury"

_children, 2021, doi:10.3390/children8110948_

Round 1

Reviewer 1 Report

This is a retrospective descriptive study using a administrative database on the prescription of nephrotoxic medication (NTM) following AKI during PICU hospitalization. The number of patients with AKI vs. those who did not have AKI is high (223 vs 747) and all diagnostic subgroups of a general PICU population are represented. To the knowledge of the authors, this is the first study to evaluate outpatient NTM prescribing following AKI in the PICU population.

The topic of this report is an important one as AKI is an significant health issue and is associated with increased morbidity and mortality in the PICU population. As mentioned in the report there are no standardized guidelines on how to follow children with AKI after hospital discharge. These patients are at risk of developing chronic kidney disease and therefor NTM prescription should be avoided.  

The manuscript is well written and offers new information. It may help to raise awareness about the importance of longterm follow-up of this patient group and the importance of NTM avoidance.

General concerns:

This report suffers the usual limitations of studies based on an administrative database. It is missing some important clinical data which limits ist ability to make robust clinical correlations as mentioned in the discussion. For example the indication for NTM was not available.

Moreover, the hospitalization dates were from 2003-2005. Since then not only prescribing parctices may have changed but also, especially in the last ten years, our understanding of AKI has dramatically increased.

Specific concerns:

INTRODUCTION: Page 1, line 33: „AKI commonly occurs due to administration of nephrotoxin medictaion“ In critically ill children, the occurence of AKI is multifactorial and NTM will be a contributing factor. However, in patients after cardiac surgery or sepsis, AKI mainly results from inadequate perfusion due to cardiopulmonary bypass and/or low blood pressure. Please add some information about the multifactorial aetiology of AKI in PICU.

MATERIAL AND METHODS. Point 2.3 AKI definition. Page 3, line 114:  

„Patients in whom AKI could not be ascertained because no SCr or UO was recorded during PICU stay were assumed to not have AKI“. The rationale behind this definition that these patients were unlikely to have been critically ill may be correct, however a level of  uncertainty remains. Maybe it would be worth mentioning in the discussion that SCr and/or UO should be documented in all patients admitted to PICU regardless of their severity of illness.

RESULTS: Table S5+6: I am suprised about the high frequency of HIV medication in this patient group. Do the authors have any explanation for this?

DISCUSSION: Page 8, line 265-270: Please double check the numbers with the result section. It seems to me that in the results section the number of patients receiving ≥3 potential NTM was 39% vs 29% and in the discussion just under 40% and 20%.

Page 9, line 303-305: Given the high incidence of ACE inhibitor prescription it would be worth discussion the value of ACE inhibitors especially in patients after cardiac surgery which represent almost 1/3 of the patients with AKI (afterload reduction vs hypertension treatment, alternatives, monitoring of renal function,…)  

Author Response

We would like to thank the reviewer for the excellent points they have brought up and hope to have satisfactorily addressed their concerns.

General concerns:

This report suffers the usual limitations of studies based on an administrative database. It is missing some important clinical data which limits ability to make robust clinical correlations as mentioned in the discussion. For example the indication for NTM was not available.

Moreover, the hospitalization dates were from 2003-2005. Since then not only prescribing parctices may have changed but also, especially in the last ten years, our understanding of AKI has dramatically increased.

RESPONSE:  We thank the reviewer for this comment and we do agree with the limitations.  We hope to have adequately highlighted these limitations in the Discussion. 

Specific concerns:

INTRODUCTION: Page 1, line 33: "AKI commonly occurs due to administration of nephrotoxin medictaion“ In critically ill children, the occurence of AKI is multifactorial and NTM will be a contributing factor. However, in patients after cardiac surgery or sepsis, AKI mainly results from inadequate perfusion due to cardiopulmonary bypass and/or low blood pressure. Please add some information about the multifactorial aetiology of AKI in PICU.

RESPONSE: Thank you for this comment. We wholeheartedly agree that the aetiology of AKI, especially in hospitalized children and in the PICU is often multifactorial and that nephrotoxic medications, while sometimes the sole cause of AKI, can also be exacerbating factors in the presence of other risk factors. We have therefore amended our initial introduction sentence to read: “While its etiology is commonly multifactorial in the hospital setting, AKI can occur and be exacerbated due to administration of nephrotoxic medications.”

MATERIAL AND METHODS. Point 2.3 AKI definition. Page 3, line 114:  "Patients in whom AKI could not be ascertained because no SCr or UO was recorded during PICU stay were assumed to not have AKI“. The rationale behind this definition that these patients were unlikely to have been critically ill may be correct, however a level of  uncertainty remains. Maybe it would be worth mentioning in the discussion that SCr and/or UO should be documented in all patients admitted to PICU regardless of their severity of illness."

RESPONSE: Thank you for this comment. We agree this is an important limitation of our study, as well as many other retrospective studies addressing pediatric AKI and one that merits specific mention. We have therefore amended a sentence in our Discussion section (5th paragraph, or second to last paragraph) to read: “Furthermore, the absence of SCr measurements in a large proportion of our cohort made identification of AKI difficult in many patients. This is a common feature of many studies of pediatric AKI and highlights how important it is to record these parameters in all children admitted to the PICU.”

RESULTS: Table S5+6: I am suprised about the high frequency of HIV medication in this patient group. Do the authors have any explanation for this?

RESPONSE: The frequency reported in Column 2 of Tables 5 and 6 represent the number of times a medication from a given drug class was recorded in our patient cohort. For example, 371 medications of the “HIV medication” class were recorded in children with AKI and 100 medications of the “HIV medication” class were recorded in children without AKI. HIV drugs were often classed individually within our database and each represent a separate count in the frequency column. However, these drugs are often given in combinations of 2 or 3 and the number of children receiving these drugs is therefore more likely to represent 30-50% of this frequency count.

DISCUSSION: Page 8, line 265-270: Please double check the numbers with the result section. It seems to me that in the results section the number of patients receiving ≥3 potential NTM was 39% vs 29% and in the discussion just under 40% and 20%.

RESPONSE: Thank you for pointing out this unfortunate typo in the Discussion on our part, which we sincerely apologize for. The results section is correct. We have modified the discussion (last sentence of the 1st Discussion paragraph) to read “Three or more potential nephrotoxic medications were prescribed to just under 40% of children with AKI and to just under 30% of children without AKI in the year following PICU discharge”

PAGE 9, LINE 303-305: Given the high incidence of ACE inhibitor prescription it would be worth discussion the value of ACE inhibitors especially in patients after cardiac surgery which represent almost 1/3 of the patients with AKI (afterload reduction vs hypertension treatment, alternatives, monitoring of renal function,…)  

RESPONSE: Thank you for this point. We have added the following sentence to our discussion (second to last sentence of the 4th paragraph of the Discussion): "ACE-inhibitors were commonly prescribed in patients with a history of AKI. This may reflect the high proportion of cardiac surgery patients in whom these medications may be indicated for afterload reduction and for the treatment of hypertension. However, considering their nephrotoxic potential, their widespread use in these at-risk populations should prompt considerations for close monitoring of kidney function.”

Reviewer 2 Report

Excellent paper, well written, methodology is very correctly described. This kind of studies do have their limitations, but have the advantage of documenting real life situation..including all patients, not having the limitations of Inclusion and exclusion criteria and the bias of informed consent procedures.

Author Response

We would like to thank the reviewer for their assessment of our manuscript and appreciate the thoughtful comments.